# Stepping forward: A study protocol for developing and validating a Malaysian diabetic foot self-care practice assessment instrument

**Divya Nair Narayanan**[ORCID]*, **Samsiah Awang**

Centre for Healthcare Quality Research, Institute for Health Systems Research, National Institutes of Health, Ministry of Health Malaysia, Setia Alam, Selangor, Malaysia

* divyanair@moh.gov.my

## Abstract

### Introduction

Diabetic foot ulcers impose significant financial burdens and diminished quality of life. Effective management relies on patients' self-care, yet often overlooked by patients as neuropathic feet are painless. In Malaysia, primary care facilities focus on prevention by promoting proper foot self-care. However, the lack of a standardised assessment tool hampers nationwide evaluation. This paper outlines the methodology for developing and validating an instrument tailored to the Malaysian context to assess diabetic foot self-care practices.

### Methods

A structured methodological process will be employed to guide the instrument's development and subsequent validation. The instrument aims to encompass comprehensible questions with a simple scoring and interpretation mechanism to foster its daily use. The study will consist of two phases. Phase 1 focuses on the development of the Malaysian instrument. This phase encompasses a literature review, item development tailored to the Malaysian context, content validity through expert panel evaluations, translations, pre-tests and a follow-up stakeholder engagement to ensure the instrument meets their requirements. Incorporating perspectives from experts and comprehensibility by local patients ensures the instrument's relevance to the local context. Phase 2 involves instrument validation through a cross-sectional study. This phase entails a pilot test and field test of the instrument among diabetic patients for validation. Cut-off ranges and their interpretations will also be established in this phase. The study sites encompass a mix of urban and rural public health clinics across Peninsular as well as East Malaysia.

provided the original author and source are credited.

**Data availability statement:** As this is a study protocol, no datasets have been generated or analysed at this stage. Upon completion and publication of the study, de-identified relevant research data will be made available upon reasonable request from the corresponding author or the Head of Centre for Biostatics & Data Repository, National Institutes of Health, Ministry of Health Malaysia, and subject to approval by the Director General of Health, Malaysia.

**Funding:** This research is funded by the Ministry of Health Malaysia Research Grant [Grant reference number: NIH/400-2/2/2 Jilid.5 (38)]. The funding body is not involved in the study design, data collection and analysis, publication decision, or manuscript preparation.

**Competing interests:** The authors have declared that no competing interests exist.

## Discussion

By developing a standard validated instrument to assess diabetic foot self-care practices, services provision gaps can be identified, and targeted interventions to improve these gaps in practices can be implemented. Individually tailored diabetes foot care education is crucial in preventing foot ulcers. This instrument can also facilitate the monitoring of improvements in patients' foot self-care practices longitudinally.

## Introduction

### Diabetes mellitus and its implications

Diabetes mellitus (DM), a silent epidemic, is a syndrome of chronic hyperglycaemia due to relative insulin deficiency, resistance or both [1]. Chronic hyperglycaemia can damage various organ systems leading to disabling and life-threatening health complications, most prominent of which are microvascular (retinopathy, nephropathy, and neuropathy) and macrovascular complications leading to an increased risk of cardiovascular diseases [2,3]. According to the Malaysian National Health Mortality and Morbidity Survey (NHMS) 2019 report, the prevalence of DM among Malaysians has increased from 13.4% in 2015 to 18.3% in 2019 [4]. The NHMS 2023 reported that 1 in 6 adult Malaysians have diabetes, of which among the adults who are aware they have diabetes, 56% do not have good blood glucose control [5].

Diabetic foot ulcer is defined as infection, ulceration, or destruction of tissues of the foot associated with peripheral neuropathy and/or peripheral arterial disease (PAD) of people with DM [6]. About 19–34% of diabetic patients develop foot ulcers at some stage in their lives [7]. Diabetic foot ulcer is a serious complication that has led to more than 80% of non-traumatic limb amputations [8] with an approximately 50% 5-year mortality [9,10]. Diabetic foot ulcers incurred an estimated cost of about RM6,000 per patient per year [11,12] in Malaysia. Besides the cost of foot complications, there are also the negative impacts relating to a loss of productivity, individual patient and family costs and loss of health-related quality of life. It poses an extensive burden to the affected individuals and the healthcare system causing socio-economic problems, consequently increasing healthcare expenditure [13,14].

### Self-care behaviours and diabetic foot management in Malaysia

Self-care behaviours are actions taken by an individual to control health problems [15], which is a basic requirement for the effectiveness of integrative treatment in chronic diseases, such as DM. Because a major part of diabetic care is undertaken by the patient, minimisation if not complete prevention of foot complications heavily depends on their self-care skills [16–19]. A previous study showed that patients with diabetes often neglect foot self-care practices due to painless neuropathic feet, which may increase the risk of diabetic foot ulcers, and concluded that these self-practices appear to be under-utilised as a primary complication prevention measure [20]. Foot self-care practices such as patient education appear to be more aggressively delivered once patients have already developed foot ulcers [20].

Although becoming accustomed to daily foot self-care such as daily foot examination can be challenging for patients with DM, these behaviours could effectively prevent foot problems by 49–85% [21], making timely foot self-management training a priority in diabetic foot ulcer prevention programs [22,23]. Prior to instructing a patient on proper foot self-care techniques, it is essential healthcare providers first understand the patient's current practices [24]. This process can be streamlined and expedited, particularly for healthcare providers managing heavy workloads, with the aid of an instrument to assess self-care practices in DM patients.

In Malaysia, DM is managed by a multi-disciplinary team, which also includes diabetic foot care. Prevention of diabetic foot ulcers is a service primarily carried out at primary care facilities. This service involves empowering patients, ensuring regular reinforcement, promoting adherence to proper foot care practices, and early recognition of ulcers. Presently, although foot self-care practices of diabetic patients are assessed; nevertheless, the assessment may be non-standardised nationwide, as there is a lack of an instrument for objective assessment. This hinders the ability to effectively compare the quality of foot self-care services offered among facilities nationwide.

### Existing instruments on diabetic foot self-care and rationale of study

A preliminary review was conducted to identify existing instruments and questionnaires used to assess diabetic foot self-care practices. This initial exploration aimed to serve as a guide and foundation for informing the development of a Malaysian instrument, to be made contextually relevant and tailored to Malaysian needs. Table 1 presents the preliminary overview findings of the various domains explored (but not limited to) in existing instruments and KAP studies. Table 2 provides a summary of selected existing instruments (but not limited to) identified from the preliminary overview, which informed the conceptual and structural considerations for the Malaysian instrument.

Existing literature has evidenced various instruments designed to assess self-care in DM patients. These instruments may either assess overall self-care, encompassing multiple components such as physical activity, medication adherence, follow-up compliance, smoking habits, diet control, foot care, and blood glucose monitoring, or focus solely on specific components [24–40]. Nevertheless, a few limitations were identified. Some of these instruments contain many items or use disaggregated component-based scoring mechanisms [24–30,33–35,39]. This may reduce both patient responses and uptake by healthcare providers due to their reduced user-friendliness and practicality for daily use. Healthcare providers often face difficulties deciding whether to assess the patient based on the total score or component subscales.

Existing instruments differ, with certain items reflecting culture-specific differences in population habits. For example, practices like daily use of socks or heating pads for the feet during cold seasons may be more relevant in Western

**Table 1. Preliminary overview – List of various domains identified from existing instruments and KAP studies [24–40].**

| DOMAINS IDENTIFIED | |
|---|---|
| Inspection of foot | Shoe to fit (feet measurement) |
| Washing of feet | Breaking into shoes |
| Drying of feet/ in between toes | Treatment of foot problems |
| Testing water temperature | Use of moisturisers/ talcums |
| Nail care | Use of socks |
| Checking inside shoes | Walking barefoot |
| Smoking affecting the health of feet | Knowing when to seek treatment |
| Importance of anti-diabetic treatment | Soaking of feet |
| Exposure of feet to heat | Types of footwear |
| Addition of irritants to water | Assigning designated time for foot care |
| Importance of foot care | Addressing wet feet |

**Table 2. Preliminary overview – Summary of existing instruments.**

| NO. | INSTRUMENT NAME | DIMENSIONS | ITEMS | SCORING | COUNTRY OF ORIGIN |
|---|---|---|---|---|---|
| 1 | Diabetic foot self-care questionnaire of the University of Malaga, Spain (DFSQ-UMA) [36] | • Foot care<br>• Shoe care<br>• Personal care | 16 | Likert | Spain |
| 2 | Nottingham Assessment of Functional Footcare (NAFF) (revised) [32] | • Foot condition<br>• Foot hygiene<br>• Shoes<br>• Foot moisturiser<br>• Nail care<br>• Prevention of foot injuries<br>• Treatment of foot injuries<br>• Use of heating items | 29 | Behavioral frequency scoring | United Kingdom |
| 3 | Diabetes foot self-care behaviour scale (DFSBS) [37] | • Foot inspection<br>• Washing<br>• Drying<br>• Toe inspection<br>• Lotion application<br>• Shoe use | 7 | Combination of Likert + Yes/No | Taiwan |
| 4 | Knowledge and practices regarding foot care in diabetic patients in Jinnah Hospital, Lahore [38] | • Compliance to medications<br>• Foot care<br>• Shoes<br>• Identification of warning signs | 15 | Yes/No | Pakistan |
| 5 | Foot Care Confidence Scale/ Foot-Care Behaviour instrument (FCCS-FCB) [39] | • Self-efficacy<br>• Preventive self-care behaviours<br>• Risky self-care behaviours | 29 | Combination of Likert + Yes/No | Mexico |
| 6 | Modified Diabetic Foot Care Knowledge and Modified Diabetic Foot Care Behaviours (MDFCK-MDFCB) [40] | • General DM management<br>• Foot condition<br>• Foot hygiene<br>• Shoes<br>• Foot moisturiser<br>• Nail care<br>• Prevention of foot injuries<br>• Treatment of foot injuries | 34 | True/False + Likert | Indonesia |

countries [24,25,27,30–33,35,36,38,39,41–44] but less relevant in Malaysia. Occupational therapists in Malaysia encountered challenges in adopting Vileikyte's instrument [25]. These include the irrelevance of certain questions, despite translation efforts, and the impracticality of domain-based scoring for daily use, which resulted in reduced uptake. Given these challenges and to better align with Malaysian stakeholder needs, it was deemed more advantageous to conduct a literature review to explore relevant evidence and subsequently develop a Malaysian instrument. This approach ensures that the Malaysian instrument is comparable to existing instruments while being tailored to the specific requirements of the local context.

A standardised and validated instrument for assessing diabetic foot self-care practices, when embraced by healthcare providers, facilitates the identification of service gaps and the implementation of individually tailored interventions to improve foot care. It can be used to track improvements in patients' foot self-care practices over time and aid in developing a formal module for diabetes-related foot care education, addressing a resource gap for occupational therapists. This instrument aims to be user-friendly for daily practice while demonstrating adequate validity and reliability. Therefore, this paper aims to outline the methodology for developing and validating a diabetic foot self-care practice assessment instrument to be used among diabetic patients referred for foot self-care services in Malaysian healthcare facilities.

## Engaging relevant stakeholders

Recognising the multifaceted nature of diabetic foot care, although the Occupational Therapy (OT) profession serves as the primary stakeholder that requested the instrument, other stakeholders such as the Family Health Development Division, Family Medicine Specialists, the Non-Communicable Disease Sector under the Disease Control Division and diabetic educators were also engaged. These stakeholders will be engaged periodically to ensure the instrument aligns with their needs. While diabetic foot care is multi-disciplinary, the instrument is intended for adoption by the OT profession for nationwide implementation in primary care clinics.

## Methodology

### Study type and design

This study will be conducted in two phases. Phase 1 involves the development of the diabetic foot self-care practice assessment instrument, and Phase 2 involves the validation of the instrument through a cross-sectional study design.

### Phase 1: Development of the Malaysian instrument

Phase 1 involves a literature review, item development tailored to better suit the Malaysian context, content validity through expert panel evaluations, translations, pre-tests and a follow-up stakeholder engagement. Fig 1 depicts the flow process.

   **a) Literature review.** A literature review will screen articles on instruments assessing diabetic foot self-care as well as knowledge, attitude and practice (KAP) questionnaires. Including KAP articles will help explore the 'practice' component, providing insights into the diverse but potentially relevant items and methods of inquiry. These insights will inform the development of the Malaysian instrument by drawing on the strengths and addressing the limitations of instruments used in other countries, while ensuring cultural and contextual relevance to the Malaysian setting. The review will also explore available scoring systems and interpretations (where available) to guide the design of an appropriate scoring framework for the Malaysian instrument. All the information extracted will be systematically tabulated to inform the development process (refer to Annex 1 in S1 File).

   This literature review will involve a search through relevant search engines such as PubMed, Scopus, Embase and CINAHL for articles and reports using the following Medical Subject Heading terms (MESH terms): "Diabetic foot" OR "diabetic feet" OR "diabetic foot ulcer" OR "diabetic foot problem" AND "self-care practice" OR "self-care behaviours" OR "self-efficacy" OR "self-management" AND "questionnaire" OR "survey" OR "instrument" OR "tool". The articles will be limited to those reported in English and Bahasa Malaysia, with no time limits.

   **b) Adaptations and modifications.** Insights from the literature review will guide the development of the proposed Malaysian instrument. The proposed instrument will be developed following the parsimonious principle [45] and utilise a Likert scale for scoring to facilitate ease of interpretation and response consistency. The Likert scale and number of items incorporated will prioritise practicality for daily use as well as ensuring usability for both healthcare professionals and patients. The design will emphasise simplicity, ease of use, and clarity, to encourage uptake by healthcare providers, accommodating their time constraints and workload. A straightforward, time-efficient instrument is also likely to improve patient cooperation. A brief section for collecting the sociodemographic profile of diabetic patients will also be developed. The proposed instrument will then be assessed by the expert panel for content validity.

   **c) Content validity of the proposed instrument.** The expert panel consisting of 5–6 experts [46] will be identified based on their area of expertise, years in service, and depending on the availability of the individual. This panel will consist of, but not limited to:

• Policymakers from the Allied Health Sciences Division

• Program planners from the Occupational Therapy Service

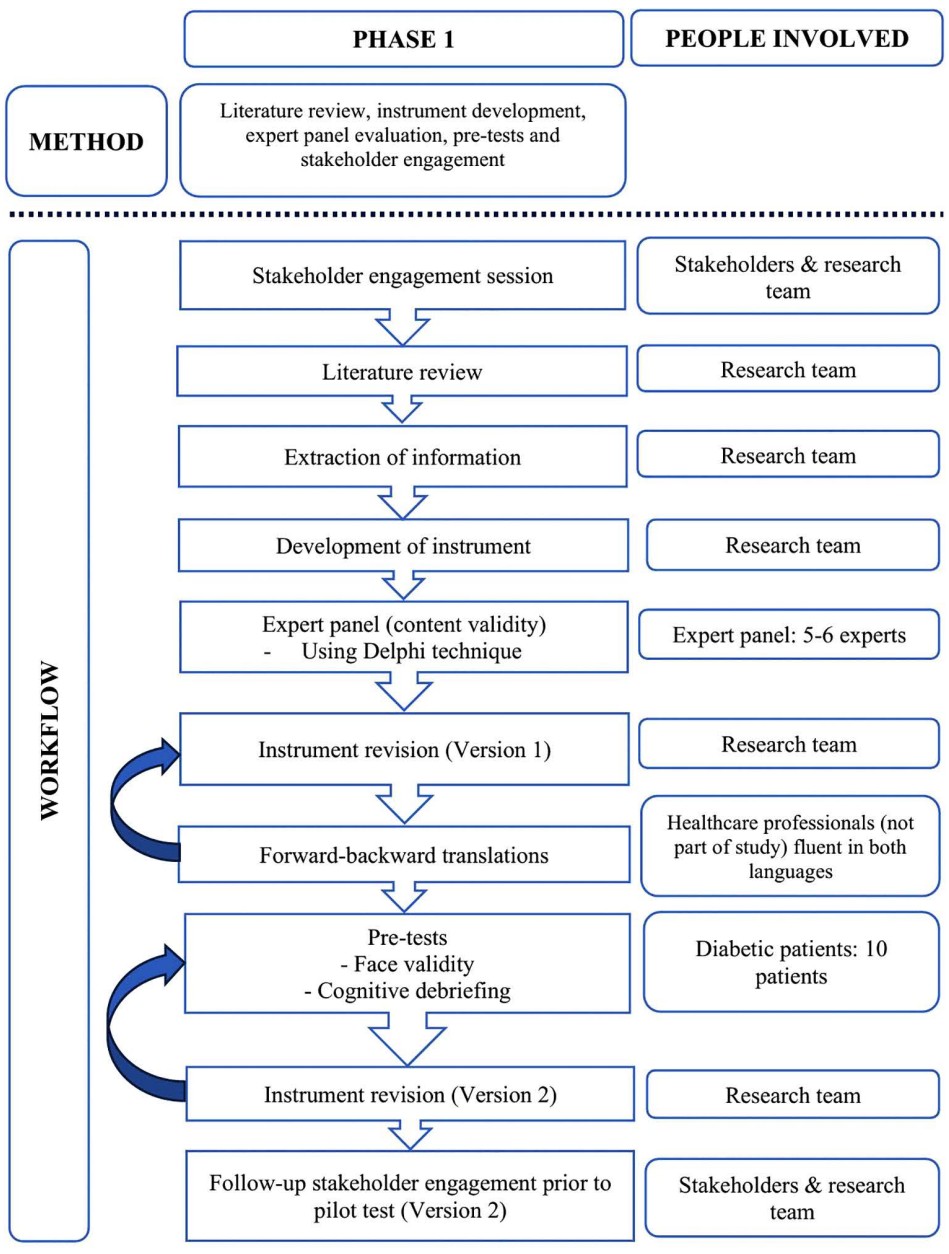

**Fig 1. Phase 1 workflow.**

- Occupational therapists managing diabetic foot care at the facility level

- Endocrinologist or Family Medicine Specialist

- Diabetic educators managing diabetic foot care at the facility level

- Other health professionals managing diabetes at the facility level

Each content expert will receive an evaluation kit comprising a cover letter, an informed consent form, the instrument consisting of the items and scoring, and the evaluation criteria form. The cover letter will outline the study's purpose, rationale for selecting the content expert, a description of the instrument, and the content evaluation procedure. The evaluation criteria form will utilise a 4-point rating scale [46] to establish the content validity.

Experts will evaluate the relevance, clarity of items, presentation, adequacy, and attainment of purpose of the instrument (refer to Annex 2 in S1 File). Feedback on the scoring mechanism will also be gathered to assess its user-friendliness in the daily Malaysian context. The experts will be requested to provide recommendations for each item they have scored low (1 or 2). They will be given two weeks to complete the evaluation and return it to the research team.

The content validity of the instrument will be established through the Content Validity Index (CVI), namely: i) Item-level CVI (I-CVI) and; ii) Scale-level CVI (S-CVI) [46]. The draft instrument and the scoring mechanism will be revised iteratively using the Delphi technique until consensus is reached [47], resulting in Version 1 of the instrument. The Delphi method is a structured process for collecting expert opinions through multiple survey rounds. After each round, feedback is provided, allowing participants to refine their responses until consensus is reached [47]. This process ensures that the instrument captures key aspects of diabetic foot self-care, considering local nuances and socio-cultural factors.

**d) Forward-backward translations of the proposed instrument.** Version 1 of the instrument will be developed in English, then translated into Bahasa Malaysia and back-translated into English to ensure the original meaning, syntax, and wording are retained. Professionals proficient in both languages and not affiliated with the study will conduct these translations. Cross-comparisons between the translator and the research team will resolve any discrepancies.

**e) Pre-test: Face validity and cognitive debriefing.** The dual-language version of the instrument will be pre-tested by occupational therapists among a group of 10 diabetic patients, who are from the intended respondent population, through purposive sampling [48]. These patients will be recruited based on age, ethnicity, gender, and education level [49].

The face validity process ensures the items are arranged meaningfully with a clear structure of the opening questions. Quantified using the Face Validity Index (FVI), this process will be assessed through a standardised form. A "yes" or "no" option will be used to evaluate the overall features, including clarity, conciseness, ease of understanding, and typographical accuracy (refer to Annex 3 in S1 File). Cognitive debriefing will be done to capture the clarity, language, and comprehensibility of the proposed items and scoring options.

The instrument will be self-administered. A consent form will be used to obtain consent from patients prior to the pre-test data collection. The patients selected for the pre-test will differ from those participating in the pilot and field tests. The responses from the pre-test will be used to further refine the instrument as an iterative process until the instrument is deemed satisfactory for the pilot test (Version 2 of the instrument). A follow-up stakeholder engagement will be conducted prior to the pilot test to ensure the instrument is aligned with their needs.

## Phase 2: Validation of the instrument

Phase 2 is a validation study which will be conducted using a cross-sectional study design. This phase involves the reliability and validity analysis as well as establishing interpretation cut-off ranges for the instrument. Fig 2 depicts the flow process of the steps in Phase 2.

**a) Pilot test.** For an instrument validation study, the pilot test assesses the preliminary feasibility of the instrument, and whether it collects all the relevant information intended to be collected [50], before deciding to conduct a full-scale field test. It can also help the research team familiarise themselves with the procedures in the instrument.

Version 2 of the instrument will be used during the pilot test. The suggested sample size for conducting a pilot test is 30–35 [51]. Thus, 30–35 adult diabetic patients will be consecutively recruited from the targeted population. Occupational therapists will collect data using the self-administered instrument. A consent form will be used to obtain participant consent prior to data collection.

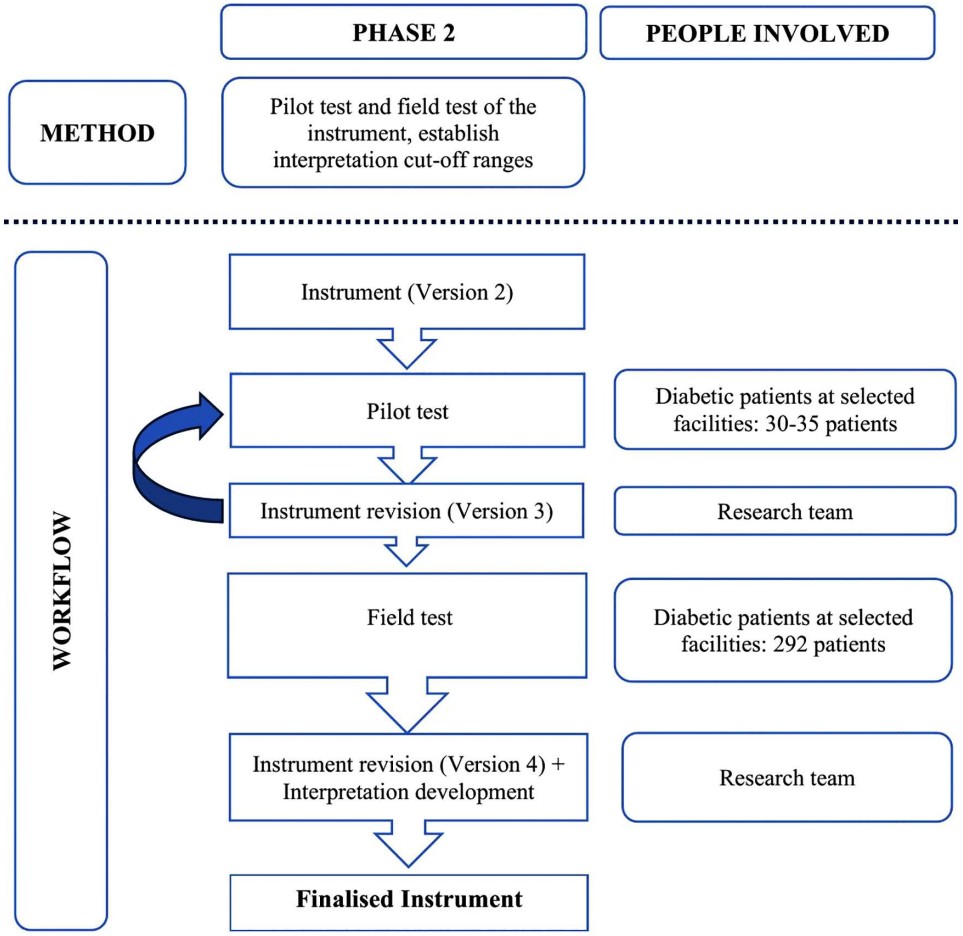

**Fig 2. Phase 2 workflow.**

Diabetic patients will complete the instrument on two separate occasions, once during their initial visit and again during a follow-up appointment. Although 1–2 weeks washout period is usually recommended [52], test-retest studies in literature use varying intervals [52]. In this study, the interval will be 1–6 weeks, aligning with the usual follow-up appointment schedule, allowing for a reasonable timeframe to assess stability in self-care behaviours while minimising patient burden. To minimise inconvenience and encourage participation, patients will be prompted to complete the instrument during their next follow-up appointment, with appointment reminders given via phone calls.

Should the reliability from the first pilot test be unsatisfactory, amendments will be made, and the pilot test will be repeated until acceptable results are achieved. If the instrument's reliability is satisfactory, the instrument (Version 3) will be used for the field test.

**b) Field test.** The field test will involve a sufficiently large sample to assess the validity of Version 3 of the instrument. Patients will be recruited through consecutive sampling to maximise the number of respondents for validation of the instrument. During the field test, dimensionality will be assessed using exploratory factor analysis (EFA), and construct validity will be determined using confirmatory factor analysis (CFA). The minimum sample size required will be calculated using the rule-of-thumb for EFA, namely the 10:1 ratio [53]. Nevertheless, for the initial estimation, at least 100 observations are required for EFA [54].

To assess CFA, the minimum sample size will be calculated through power analysis using G*Power software. The DFSQ-UMA [36], a validated 16-item instrument used in Spain, will be utilised to calculate the estimated required sample size for CFA. With a medium effect size, an alpha of 0.05, a power of 0.8, and 16 predictors (based on the DFSQ-UMA), the estimated minimum sample size is 143. Accounting for a 20% non-response rate, the preliminary estimate suggests a minimum of 292 participants will be necessary for both EFA and CFA. Nevertheless, the final required sample size for the field test will depend on the number of items included in the proposed Malaysian instrument.

Occupational therapists will collect data using the self-administered instrument. A consent form will be used to obtain participant consent prior to data collection. During the field test, the time taken by patients to complete the instrument will also be measured.

Developing meaningful thresholds to aid the interpretation of patient-reported outcome measures (PROMs) has become standard practice due to the growing emphasis on patient-centered care [55]. Analysing field test data across age groups, major ethnicities, and urban/rural areas will allow for comparisons before setting standardised threshold scores for national interpretation. These thresholds aim to categorise diabetic patients into specific groups, such as poor, moderate, and good. This approach enhances precision and sensitivity, accurately reflecting the varying levels of self-care practices. These refined categories will support more informed decision-making for improving patients' foot self-care practices.

## Study population and sampling

The study population includes any diabetic patient referred for foot care in selected Ministry of Health primary care health clinics, as listed. The pilot and field tests will be conducted at occupational therapy departments of government primary care health clinics in selected states (based on regions, i.e., northern, southern, central, east-coast, Sabah, and Sarawak). Both urban and rural settings were factored in selecting study sites. Clinics were first purposively selected based on the availability of a permanent occupational therapist, followed by ensuring the clinics cater to an adequate patient population to meet the required sample size.

In total, seven clinics were chosen as study sites. Five clinics (two urban, three rural) were selected as pilot test sites, while seven clinics (four urban, three rural), which included the pilot test sites, were selected as field test sites. These study sites were chosen for their high patient volume. Table 3 lists the proposed data collection sites.

For the five clinics listed as both pilot and field test sites, different diabetic patients will be recruited for each test. Occupational therapists at these sites will maintain separate databases for pilot and field test patients, with patient identification numbers as identifiers to prevent patient overlap. Responses will be de-identified before submission to the research team for analysis.

A training session on the instrument will be conducted prior to the pilot and field tests. This session will assist occupational therapists from the study sites in familiarising themselves with the instrument, ensuring a standardised understanding of the procedure, instrument, and technical specifications. To support application post-training, a quick-reference training guidebook with a flowchart outlining the data collection process will also be developed and provided to study site

Table 3. List of proposed data collection sites.

| NO. | STATE | REGION | CLINIC SETTING | AREA | TYPE OF TEST |
|---|---|---|---|---|---|
| 1. | Perlis | Northern | Kampung Gial Health Clinic | Rural | Pilot & field test |
| 2. | Selangor | Central | Rasa Health Clinic | Rural | Pilot & field test |
| 3. | Kuala Lumpur | Central | Kuala Lumpur Health Clinic | Urban | Field test |
| 4. | Terengganu | East Coast | Manir Health Clinic | Urban | Pilot & field test |
| 5. | Johor | Southern | Sungai Mati Health Clinic | Rural | Pilot & field test |
| 6. | Sabah | East Malaysia | Luyang Health Clinic | Urban | Field test |
| 7. | Sarawak | East Malaysia | Jalan Masjid Health Clinic | Urban | Pilot & field test |

investigators as reference materials. Fig 3 depicts the data collection flowchart, as detailed in the training guidebook. Data collected will then be submitted to the research team for entry and analysis.

The study will include patients diagnosed with Type II diabetes mellitus who are Malaysian citizens aged 18 and above. Patients will be excluded if they have bilateral lower limb amputations, cognitive impairments, are dependent on

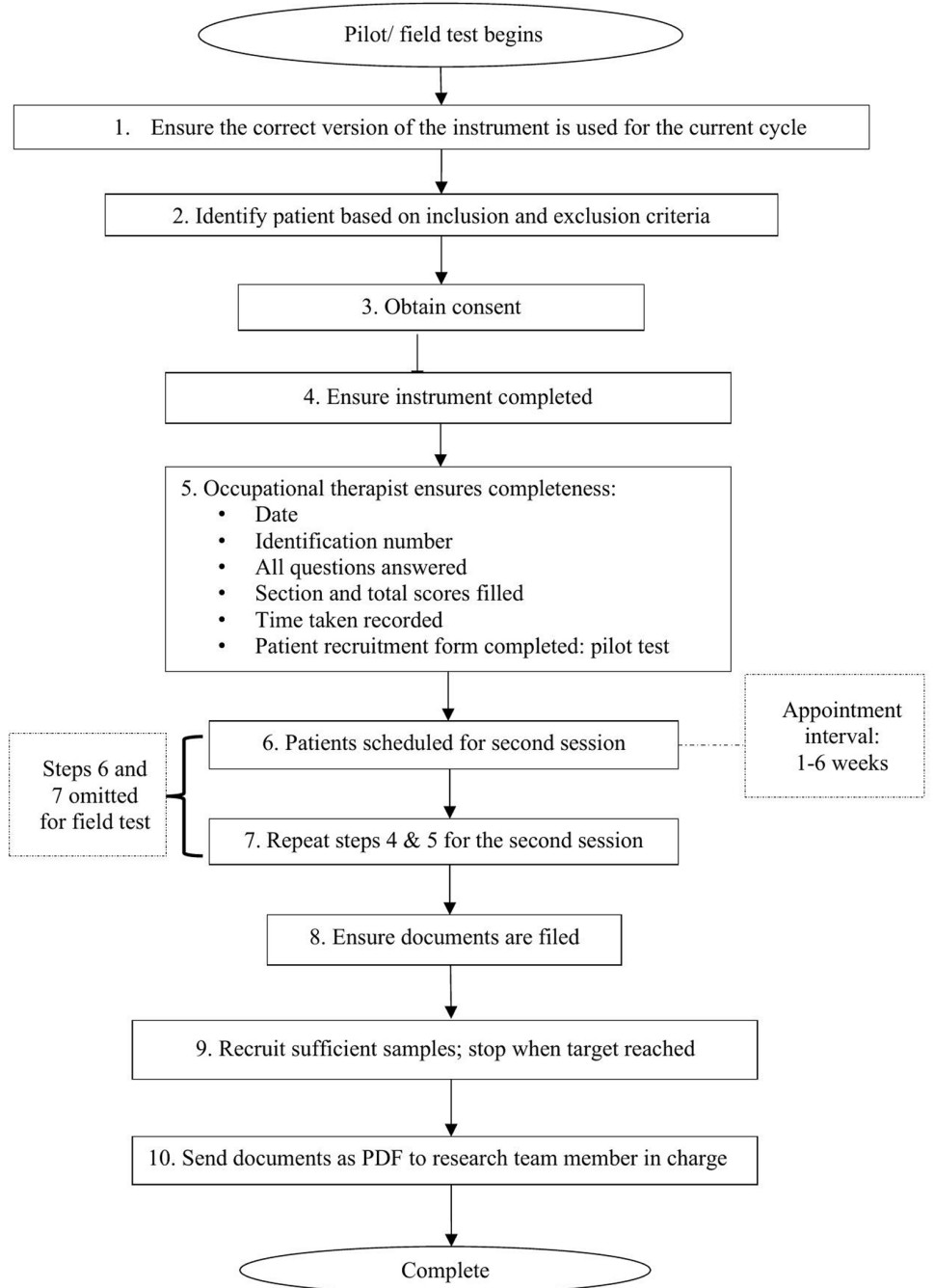

**Fig 3. Data collection flowchart.**

caregivers for daily activities, or are not proficient in either Bahasa Malaysia or English. Patients may withdraw from the study at any time without penalty or consequence. They may continue their regular foot care follow-ups as usual. However, to maintain the required sample size, any withdrawn patients will be replaced.

### Ethical approval and study registration

Participation in the study is entirely voluntary, and all recruited participants will provide written informed consent. They will be explicitly informed of the study's nature, purpose, objectives, confidentiality and anonymity assurances, as well as their right to withdraw from the study at any time, should they wish to do so. No personal information will be disclosed when the findings of the study are published. The study will adhere to the ethical principles outlined in the Declaration of Helsinki and the Malaysian Good Clinical Practice Guidelines. It is registered with the National Medical Research Register (NMRR ID-24-00563-1ZX), and ethical approval has been granted by the Medical Research and Ethics Committee, Ministry of Health (ID-24-00563-1ZX). The study duration spans from August 2024 to April 2026. The pilot and field tests, during which patients will be recruited, are expected to begin in June 2025 and be completed by October/November 2025. The study is expected to be completed by April 2026.

### Data management and oversight

Access to the data will be limited to the research team members directly involved in data analysis and processing. Strict confidentiality measures will be followed to protect the patients' identities and sensitive information. All data will be anonymised, and unique identifiers will be assigned to patients to ensure confidentiality during analysis and reporting. The final refined version of the collected data will be stored in a designated repository in the Ministry of Health for seven years after the completion of the research.

### Statistical analysis plan

All collected data will be subjected to regular quality checks, with the research team divided into small groups assigned to each study site to independently verify data entries. This process will ensure discrepancies are identified and resolved promptly, maintain completeness and accuracy in data entry. Statistical analyses will be performed using the Statistical Package for the Social Sciences (SPSS) Software (Version 26). Descriptive analysis will be used to describe the sociodemographic profiles.

   **a) Content validity & face validity.**  The content validity will be estimated quantitatively using the CVI based on the expert panel evaluations (refer to Annex 4 in S1 File). A CVI of at least 0.83 will be considered acceptable [46]. Face validity will be estimated using FVI via a structured form (refer to Annex 4 in S1 File), wherein an FVI of at least 0.83 will be deemed acceptable [48]. Amendments will be made based on the feedback received.

   **b) Reliability.**  Inter-item correlations between 0.15 and 0.5 will be deemed acceptable [56]. An item-total correlation of ≥ 0.3 will indicate that an item is related to the overall scale [57]. Cronbach's α values of 0.7 or above will be considered acceptable [58]. Test-retest reliability will be determined using Intraclass Correlation Coefficient (ICC), with a value greater than 0.75 [59] demonstrating a good level of agreement.

   **c) Exploratory factor analysis.**  The Kaiser-Meyer-Olkin (KMO) criterion will be assessed for sampling adequacy, with a value of ≥ 0.50 considered good [60]. A significant Chi-square value from Bartlett's test of sphericity will indicate that relationships exist in the data, and at least one domain is present [57]. Domains will be extracted using principal axis factoring and according to the Kaiser criterion, domains with eigenvalues of ≥ 1.0 should be retained [61]. Following domain extraction, Promax rotation will be applied, and factor loadings of > 0.7 is desirable, with a minimum threshold of 0.4 [54].

   **d) Confirmatory factor analysis (CFA).**  When assessing the convergent validity, factor loadings and the average variance extracted (AVE) will be examined. Loading values ≥ 0.5 are acceptable, and high factor loading scores that

contribute to AVE scores of greater than 0.5 will be deemed valid [62]. If convergent validity is not acceptable, items with low factor loadings will be reviewed and revised or removed as needed. Discriminant validity will be confirmed if the square root of each construct's AVE exceeds its correlations with other latent constructs, as per the Fornell-Larcker criterion [63]. If discriminant validity is not acceptable, constructs showing significant overlap will be redefined, and items contributing to cross-loadings will be revised or excluded.

## Discussion

This article outlines the methodology for the development and validation of a Malaysian diabetic foot self-care assessment instrument, addressing the current gap in locally suited instruments that are practical for daily use in our healthcare system. This article could serve as a guide for other Asian countries facing similar challenges, enabling them to develop their own instruments without concerns about proprietary restrictions. While several internationally available instruments exist for assessing diabetic foot self-care, developing a local instrument offers distinct advantages by catering to the unique needs of the Malaysian healthcare system and its population.

This study's short-term benefits lie in its ability to improve the diabetic foot care services provided to patients. International instruments may overlook local practices such as crossing legs and sitting on the floor, which commonly occurs during prayers, religious sermons, and certain cultural settings where meals are served while sitting on the floor. They may also miss practices such as washing of feet before prayers, while including less relevant practices such as the wearing of boots. A locally developed instrument tailored to Malaysia's cultural and healthcare system capacity would make self-care assessments more understandable and resonate better with patients. By aligning with Malaysian lifestyles, this instrument allows healthcare providers to identify and address risk factors unique to Malaysia, such as foot care habits influenced by local cultural practices and traditional medicine. This approach will enable earlier interventions and more targeted support for high-risk patients.

In the medium and long term, integrating this instrument into patient management can potentially reduce diabetic foot complications such as amputations and hospitalisations, through early identification and intervention. This reduction would alleviate the strain on healthcare resources, leading to a more efficient and sustainable healthcare system, while improving patients' quality of life. Guided by the instrument, healthcare providers can tailor education plans specific to patients, addressing patient-specific issues, ensuring patient receives care that is relevant and effective, which aims to improve self-care practices, thereby improving outcomes. The instrument looks from the self-care perspective aiming to empower the community to take an active role in managing and preventing diabetes-related complications. This increased health literacy will foster a healthier and more informed population.

The adoption of this locally developed instrument will also support the standardisation of diabetic foot care assessments across Malaysia and promote uniformity in clinical practice at the national level. This standardisation will facilitate benchmarking and continuous improvement, as healthcare providers can track progress and outcomes with greater precision. Beyond individual care, with nationwide uptake and consistent use of the instrument, a database of long-term trends in diabetic foot care can be generated. By providing policymakers with long-term, context-specific evidence, this approach can improve existing evidence-based guidelines and support research on innovative patient education and early intervention strategies tailored to the unique needs of Malaysian patients. It can also inform resource allocation for public health initiatives ensuring that patients receive care that is both relevant and effective.

While this study has garnered support from relevant stakeholders, nevertheless, development of this instrument may still face several challenges. Firstly, ensuring the instrument is culturally sensitive and inclusive, reflecting Malaysia's diverse population, though difficult, is crucial. Secondly, limited resources and time constraints may impact the study's progress, while retaining a representative sample could prove challenging. Thirdly, the variability in healthcare infrastructure between urban and rural areas requires the instrument to be adaptable and effective across different settings. Finally, ensuring the instrument's long-term sustainability and scalability is another challenge, ensuring it remains relevant and updated as practices and needs evolve.

## Conclusion

Ultimately, while international instruments provide a general framework, a local instrument offers the distinct advantage of cultural relevance, better patient engagement, and the ability to address specific challenges faced by the Malaysian population. Developing a local instrument can contribute to significant improvements in patient care and long-term advancements in public health, self-empowerment, research, and healthcare policy.

## Supporting information

**S1 File. All tables are provided in Annexes 1–4.**
(PDF)

## Acknowledgments

The authors would like to thank the Director-General of Health Malaysia for his approval to publish this work. We would like to thank all research team members and stakeholders for their cooperation and contributions.

## Author contributions

**Conceptualization:** Divya Nair Narayanan, Samsiah Awang.

**Data curation:** Divya Nair Narayanan, Samsiah Awang.

**Formal analysis:** Divya Nair Narayanan.

**Funding acquisition:** Divya Nair Narayanan, Samsiah Awang.

**Methodology:** Divya Nair Narayanan.

**Project administration:** Divya Nair Narayanan.

**Supervision:** Samsiah Awang.

**Writing – original draft:** Divya Nair Narayanan.

**Writing – review & editing:** Divya Nair Narayanan, Samsiah Awang.

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
