## [Decision Letter · Decision Letter 0]

5 May 2025

Dear Dr. Nair Narayanan,

Thank you for submitting your manuscript to PLOS ONE. After careful consideration, we feel that it has merit but does not fully meet PLOS ONE’s publication criteria as it currently stands. Therefore, we invite you to submit a revised version of the manuscript that addresses the points raised during the review process.

We look forward to receiving your revised manuscript.

Kind regards,

Yee Gary Ang, MBBS MPH

Academic Editor

PLOS ONE

Journal Requirements:

5. In the online submission form, you indicated that data used in this study is not publicly available due to confidentiality restrictions. However, data can be accessed upon reasonable request and subject to approval from the corresponding author or the Head of Centre for Biostatics & Data Repository, National Institutes of Health, Ministry of Health Malaysia on reasonable grounds for the request and with the permission from the Director General of Health, Malaysia.

Reviewers' comments:

Reviewer's Responses to Questions

1. Does the manuscript provide a valid rationale for the proposed study, with clearly identified and justified research questions?

Reviewer #1: Yes

2. Is the protocol technically sound and planned in a manner that will lead to a meaningful outcome and allow testing the stated hypotheses?

Reviewer #1: Partly

3. Is the methodology feasible and described in sufficient detail to allow the work to be replicable?

Reviewer #1: No

4. Have the authors described where all data underlying the findings will be made available when the study is complete?

Reviewer #1: No

5. Is the manuscript presented in an intelligible fashion and written in standard English?

Reviewer #1: Yes

You may also provide optional suggestions and comments to authors that they might find helpful in planning their study.

Reviewer #1: I see ethical problems with the use of items from other global scales and, worse, with the adaptation and modification of items from these scales without prior authorization from the authors of the original scales.

I suggest taking the best scale and, with the authorization of the authors of the original scale, translating it into the language of your country. I recommend TO REFUSE to publish this article in its current form.

Do you want your identity to be public for this peer review? For information about this choice, including consent withdrawal, please see our Privacy Policy

Reviewer #1: Yes: THEREZA MARIA MAGALHÃES MOREIRA

---

## [Author Response · Author response to Decision Letter 1]

19 May 2025

The PLOS ONE style templates can be found at https://journals.plos.org/plosone/s/file?id=wjVg/PLOSOne_formatting_sample_main_body.pdf and https://journals.plos.org/plosone/s/file?id=ba62/PLOSOne_formatting_sample_title_authors_affiliations.pdf

Response: Thank you for the guidance. Where applicable and required, the manuscript has been reformatted to align with the PLOS ONE style requirements.

Response: Thank you for highlighting the discrepancy.

• I have amended the ‘Funding Information’ as instructed to ensure it is accurate to reflect the following:

This research is funded by the Ministry of Health Malaysia Research Grant [Grant reference number: KKM.400-4/2/30 Jld.2 (52)]. The funding body is not involved in the study design, data collection and analysis, publication decision, or manuscript preparation.

• Nevertheless, I am unable to amend the 'Financial Disclosure' section, as I no longer have access to it, possibly because the journal has not granted permission for further edits to that section.

Response: Thank you for highlighting this important aspect.

• We have now ensured that the full ethics statement is included exclusively in the “Methods” section (lines 377–389: marked up version), as advised.

4. We note that you have indicated that there are restrictions to data sharing for this study. For studies involving human research participant data or other sensitive data, we encourage authors to share de-identified or anonymized data. However, when data cannot be publicly shared for ethical reasons, we allow authors to make their data sets available upon request. For information on unacceptable data access restrictions, please see http://journals.plos.org/plosone/s/data-availability#loc-unacceptable-data-access-restrictions. Before we proceed with your manuscript, please address the following prompts:

a) If there are ethical or legal restrictions on sharing a de-identified data set, please explain them in detail (e.g., data contain potentially identifying or sensitive patient information, data are owned by a third-party organization, etc.) and who has imposed them (e.g., a Research Ethics Committee or Institutional Review Board, etc.).

b) Please also provide contact information for a data access committee, ethics committee, or other institutional body to which data requests may be sent.

c) If there are no restrictions, please upload the minimal anonymized data set necessary to replicate your study findings to a stable, public repository and provide us with the relevant URLs, DOIs, or accession numbers.

d) Please see http://www.bmj.com/content/340/bmj.c181.long for guidelines on how to de-identify and prepare clinical data for publication. For a list of recommended repositories, please see https://journals.plos.org/plosone/s/recommended-repositories.

e) You also have the option of uploading the data as Supporting Information files, but we would recommend depositing data directly to a data repository if possible.

Response: Thank you for your detailed guidance on data sharing requirements.

• As this is a study protocol, no data have been generated or analysed at this stage. However, we acknowledge the importance of data availability and have updated the “Data Availability Statement” section in the manuscript and submission form to reflect the following:

• As this is a study protocol, no datasets have been generated or analysed at this stage. Upon completion and publication of the study, de-identified relevant research data will be made available upon reasonable request from the corresponding author or the Head of Centre for Biostatics & Data Repository, National Institutes of Health, Ministry of Health Malaysia, and subject to approval by the Director General of Health, Malaysia. (as elaborated from lines 370-376: marked up version)

5. Reviewer #1: I see ethical problems with the use of items from other global scales and, worse, with the adaptation and modification of items from these scales without prior authorization from the authors of the original scales. I suggest taking the best scale and, with the authorization of the authors of the original scale, translating it into the language of your country. I recommend TO REFUSE to publish this article in its current form.

Response:

Thank you for your valuable feedback. We understand and appreciate your concerns regarding the use and adaptation of existing global instruments.

• In response, we have carefully reconsidered our approach and have revised the protocol accordingly.

• The proposed study will no longer involve the adoption or adaptation of any existing scales; instead, we intend to develop a new Malaysian-specific instrument.

• To inform/guide this development, the comprehensive literature review will examine the strengths and limitations of previously established instruments.

• Insights gained from this review will guide the design of our instrument, aligning it with the specific needs of Malaysian stakeholders while also ensuring internationally recognised constructs to support cross-cultural comparability.

• We are also mindful of potential proprietary and licensing issues that may arise from using established tools. For example, the Morisky Medication Adherence Scale, which was previously adopted and adapted in our country, later required licensing fees that became too costly for sustained national use and was eventually discontinued.

• Developing a contextually relevant instrument from the outset will help us avoid such challenges and ensure broader accessibility for future nationwide implementation.

• We remain committed to ethical research practices and will ensure that content drawn from existing instruments are appropriately acknowledged and cited.

• Additionally, we have expanded the methodological section to provide greater detail on our instrument development process. This is intended to enhance transparency and support replicability of the study by researchers in other countries with similar contextual needs.

• Amended in manuscript (marked up version) in the following lines:

28-31

92-95

97-99

120-125

147-151

158-165

174-178

256-257

312-315

321-322

326-328

391-392

421-423

6. Have the authors described where all data underlying the findings will be made available when the study is complete?

Reviewer #1: No

Response: Thank you for highlighting this important point.

• As this is a study protocol, no datasets have been generated or analysed at this stage.

• However, we have revised the manuscript under the “Data Availability Statement” clarifying that, upon completion and publication of the study, de-identified relevant data will be made available upon reasonable request from the corresponding author or the Head of the Centre for Biostatistics & Data Repository, National Institutes of Health, Ministry of Health Malaysia, and subject to approval by the permission from the Director General of Health, Malaysia. (as elaborated from lines 370-376: marked up version)

7. While revising your submission, please upload your figure files to the Preflight Analysis and Conversion Engine (PACE) digital diagnostic tool, https://pacev2.apexcovantage.com/. PACE helps ensure that figures meet PLOS requirements.

Response: Thank you for the instructions.

• All figure files have been uploaded to the Preflight Analysis and Conversion Engine (PACE) and have passed the required checks.

• We have ensured that the figures meet PLOS formatting requirements.

• Fig 1, Fig 2 and Fig 3 have been uploaded as .tif files, labelled accordingly (label of figure).

---

## [Decision Letter · Decision Letter 1]

2 Jun 2025

Dear Dr. Nair Narayanan,

Thank you for submitting your manuscript to PLOS ONE. After careful consideration, we feel that it has merit but does not fully meet PLOS ONE’s publication criteria as it currently stands. Therefore, we invite you to submit a revised version of the manuscript that addresses the points raised during the review process.

**We only managed to find one reviewer**

**Please make the changes and consider resubmitting**

We look forward to receiving your revised manuscript.

Kind regards,

Yee Gary Ang, MBBS MPH

Academic Editor

PLOS ONE

Journal Requirements:

Reviewers' comments:

Reviewer's Responses to Questions

**Comments to the Author**

1. Does the manuscript provide a valid rationale for the proposed study, with clearly identified and justified research questions?

Reviewer #2: Yes

2. Is the protocol technically sound and planned in a manner that will lead to a meaningful outcome and allow testing the stated hypotheses?

Reviewer #2: Partly

3. Is the methodology feasible and described in sufficient detail to allow the work to be replicable?

Reviewer #2: Yes

4. Have the authors described where all data underlying the findings will be made available when the study is complete?

Reviewer #2: Yes

5. Is the manuscript presented in an intelligible fashion and written in standard English?

Reviewer #2: Yes

You may also provide optional suggestions and comments to authors that they might find helpful in planning their study.

Reviewer #2: The aim of this study protocol is describe the development and validation of a Malaysian diabetic foot self-care practice assessment instrument.

In this revised version, the recommendations previously made have been taken into account and the manuscript has been improved, and I have only some minor comments.

1) At the end of the introduction, the aim and/or objectives of the study should be clearly defined. As the proposed study will first develop and then validate a national instrument, both aspects should also be mentioned in the objectives. As the authors have changed the structure of the study compared to the first version of the manuscript, they should also change the title.

2) Study type and design – please clearly state that the first stage is the development of the instrument and the second is the validation study.

3) Patients will be involved in the second phase of the study - validation through a cross-sectional study. However, the new instrument will be self-administered by the patients, so the patients should be involved already in the development process - from the definition of the topics/items of the instrument and the content validation, patients should be involved in the expert group, in addition it may be appropriate to conduct a qualitative survey among patients.

**Do you want your identity to be public for this peer review?** For information about this choice, including consent withdrawal, please see our Privacy Policy

Reviewer #2: No

---

## [Author Response · Author response to Decision Letter 2]

17 Jul 2025

Journal Requirements / Editor’s Comments:

1. Your ethics statement should only appear in the Methods section of your manuscript. If your ethics statement is written in any section besides the Methods, please delete it from

any other section.

Response:

• Thank you for your attention to this detail.

• I have carefully reviewed the entire manuscript to ensure that the ethics statement appears only in the Methods section.

• The manuscript’s main sections are: Abstract, Introduction, Methods, Discussion, Conclusion, Acknowledgment, Supporting Information, and References.

• Within the Methods section, the subheadings are: Study Type & Design, Phase 1: Development of the Malaysian Instrument, Phase 2: Validation of the Instrument, Study

Population and Sampling, Statistical Analysis Plan, Data Management and Oversight, Data Availability Statement, and Ethical Approval & Study Registration.

• I confirm that the ethics statement appears only under the Ethical Approval & Study Registration subheading within the Methods section and does not appear elsewhere in the

manuscript.

2.

• Please review your reference list to ensure that it is complete and correct.

• If you have cited papers that have been retracted, please include the rationale for doing so in the manuscript text, or remove these references and replace them with relevant

current references.

• Any changes to the reference list should be mentioned in the rebuttal letter that accompanies your revised manuscript.

• If you need to cite a retracted article, indicate the article’s retracted status in the References list and also include a citation and full reference for the retraction notice.

Response:

• We thank the reviewer for this important reminder.

• In response, we undertook a thorough review of the entire reference list to ensure that every source cited is complete, verifiable, and fully compliant with PLOS ONE’s rigorous

publication standards.

• We confirm there were no retracted articles in the reference list.

• Nevertheless, 3 references were removed as they may not meet the required standards for robust peer-review, retrievability, or scientific credibility:

a. Goweda et al. (2017) was removed because it is not widely indexed and its content is sufficiently covered by stronger, peer-reviewed sources already included in the

manuscript.

b. AHS et al. (2018) was removed due to lack of indexing and unclear peer-review status.

- To uphold the scientific foundation of this section, we actively replaced it with a more credible, peer-reviewed source (Sulistyo et al., 2018, Journal of Research in Nursing),

which reports on the same instrument and context.

c. Quadri et al. (2013) was removed as it is a conference abstract only, with no peer-reviewed full-text version available, and therefore does not meet PLOS ONE’s citation

requirements.

• By carefully verifying, removing, and replacing these references, we have taken deliberate steps to strengthen the scholarly rigor and integrity of the manuscript.

• We have ensured that all references meet the journal’s expectations for transparency, retrievability, and quality.

Reviewer’s Comments:

1. Is the protocol technically sound and planned in a manner that will lead to a meaningful outcome and allow testing the stated hypotheses?

Reviewer #2: Partly

Response:

• We have addressed these methodological aspects through our responses to the specific comments provided and have revised the manuscript accordingly to ensure sufficient

detail, clarity, and transparency.

2. Review Comments to the Author: The aim of this study protocol is describe the development and validation of a Malaysian diabetic foot self-care practice assessment instrument.

In this revised version, the recommendations previously made have been taken into account and the manuscript has been improved, and I have only some minor comments.

Response:

• We thank the reviewer for acknowledging the improvements and appreciate the constructive feedback provided throughout the review process.

• We will address the remaining minor comments accordingly to further strengthen the manuscript.

3. Review Comments to the Author:

• At the end of the introduction, the aim and/or objectives of the study should be clearly defined.

• As the proposed study will first develop and then validate a national instrument, both aspects should also be mentioned in the objectives.

Response:

• We thank the reviewer for this valuable suggestion.

• In response, we have revised the study objectives in both the Abstract and Introduction sections to explicitly state that the study involves both the development and validation

of the national instrument.

- Lines 23, 25, 26 (Abstract)

- Line 130 (Introduction)

• We believe this amendment clarifies the full scope of the study as intended.

4. Review Comments to the Author: As the authors have changed the structure of the study compared to the first version of the manuscript, they should also change the title.

Response:

• We thank the reviewer for this constructive suggestion. We concur that the revised structure of the study should be clearly reflected in the title.

• Accordingly, we have amended the title to: “Stepping Forward: A Study Protocol for Developing and Validating a Malaysian Diabetic Foot Self-care Practice Assessment

Instrument” and the short title to “Developing and validating a Malaysian diabetic foot self-care practice assessment instrument”

• These revised titles more accurately represent the comprehensive scope of the study, which now encompasses both the development and validation phases of the proposed

national instrument.

5. Review Comments to the Author: Study type and design - please clearly state that the first stage is the development of the instrument and the second is the validation study.

Response:

• We thank the reviewer for this helpful suggestion.

• We have revised the manuscript to clearly state that Phase 1 involves the development of the instrument and Phase 2 involves its validation through a cross-sectional study

design.

- Lines 144-146

- Lines 383-386

6. Review Comments to the Author:

• Patients will be involved in the second phase of the study - validation through a cross-sectional study.

• However, the new instrument will be self-administered by the patients, so the patients should be involved already in the development process - from the definition of the

topics/items of the instrument and the content validation, patients should be involved in the expert group, in addition it may be appropriate to conduct a qualitative survey

among patients.

Response:

• We sincerely thank the reviewer for highlighting this important point.

• We agree that incorporating patient perspectives is crucial to ensure that the instrument is clear, relevant, and practical for its intended users.

• In Phase 1, the domains and items will be systematically drafted based on established national diabetic foot care guidelines and robust international evidence.

• This initial version will undergo expert content validation by a panel of healthcare professionals with extensive experience in diabetic foot care to ensure clinical relevance and

accuracy.

• To ensure that the instrument also meets patient needs and expectations, patients will be involved in the face validity process in Phase 1 through pre-testing and cognitive

debriefing.

• This will include:

- A short, structured feedback form for patients

- Cognitive debriefing interviews to gather in-depth feedback on wording, cultural appropriateness, and ease of use

• We will carefully analyse this feedback to refine the instrument before proceeding to piloting and field-testing in Phase 2 : Lines 216–232

• This multi-step approach ensures that the instrument content is evidence-based, clinically sound, and meaningfully co-shaped by patient input, hence maximizing its practical

usability and cultural appropriateness.

---

## [Decision Letter · Decision Letter 2]

14 Nov 2025

Stepping forward: A study protocol for developing and validating a Malaysian diabetic foot self-care practice assessment instrument

PONE-D-25-16530R2

Dear Dr. Nair Narayanan,

We’re pleased to inform you that your manuscript has been judged scientifically suitable for publication and will be formally accepted for publication once it meets all outstanding technical requirements.

Kind regards,

Yee Gary Ang, MBBS MPH

Academic Editor

PLOS ONE

Additional Editor Comments (optional):

Reviewers' comments:

Reviewer's Responses to Questions

**Comments to the Author**

1. Does the manuscript provide a valid rationale for the proposed study, with clearly identified and justified research questions?

Reviewer #2: Yes

Reviewer #3: Yes

2. Is the protocol technically sound and planned in a manner that will lead to a meaningful outcome and allow testing the stated hypotheses?

Reviewer #2: Yes

Reviewer #3: Yes

3. Is the methodology feasible and described in sufficient detail to allow the work to be replicable?

Reviewer #2: Yes

Reviewer #3: Yes

4. Have the authors described where all data underlying the findings will be made available when the study is complete?

Reviewer #2: Yes

Reviewer #3: Yes

5. Is the manuscript presented in an intelligible fashion and written in standard English?

Reviewer #2: Yes

Reviewer #3: Yes

You may also provide optional suggestions and comments to authors that they might find helpful in planning their study.

Reviewer #2: In this revised version, the recommendations previously made have been taken into account and the manuscript has been improved, and I have no further suggestions.

Reviewer #3: The revision is clear and practical, and the steps are easy to follow for replication. The study idea fits well with current work in this area. I don’t see major problems. I think this protocol can be accepted for publication.

**Do you want your identity to be public for this peer review?** For information about this choice, including consent withdrawal, please see our Privacy Policy

Reviewer #2: **Yes: ** Kaja Põlluste

Reviewer #3: No

---

## [Editor Report · Acceptance letter]

PONE-D-25-16530R2

PLOS One

Dear Dr. Nair Narayanan,

I'm pleased to inform you that your manuscript has been deemed suitable for publication in PLOS One. Congratulations! Your manuscript is now being handed over to our production team.

Kind regards,

on behalf of

Dr. Yee Gary Ang

Academic Editor

PLOS One